# The Risk of SARS-CoV-2 Infection in Pregnant Women: An Observational Cohort Study Using the BIFAP Database

**DOI:** 10.3390/healthcare10122429

**Published:** 2022-12-02

**Authors:** Mercedes Mota, Consuelo Huerta-Álvarez, Ana Llorente, Lucia Cea-Soriano

**Affiliations:** 1Department of Public Health and Maternal Child Health, Faculty of Medicine, Complutense University of Madrid, 28040 Madrid, Spain; 2BIFAP, Division of Pharmacoepidemiology and Pharmacovigilance, Spanish Agency for Medicines and Medical Devices (AEMPS), 28022 Madrid, Spain

**Keywords:** pregnancy, SARS-CoV-2, cohort study, incidence rates

## Abstract

Background: It has been suggested that women experiencing during pregnancy several physiological and immunological changes that might increase the risk of any infection including the SARS-CoV-2. Objective: We aimed to quantify the risk of SARS-CoV-2 infection during pregnancy compared with women with no pregnancies. Methods: We used data from the BIFAP database and a published algorithm to identify all pregnancies during 2020. Pregnancies were matched (1:4) by age region, and length of pregnancy with a cohort of women of childbearing age. All women with SARS-CoV-2 infection before entering the study were discarded. We estimated incidence rates of SARS-CoV-2 with 95% confidence intervals (CIs) expressed by 1000 person-months as well as Kaplan–Meier figures overall and also stratified according to pregnancy period: during pregnancy, at puerperium (from end of pregnancy up to 42 days) and after pregnancy. (from 43 days after pregnancy up to end pf study period (i.e., June 2021). We conducted a Cox regression to assess risk factors for SARS-COV infection. The incidence rate of SARS-CoV-2 infection expressed by 1000 person-months were. Results: There was a total of 103,185 pregnancies and 412,740 matched women at childbearing, with a mean age of 32.3 years. The corresponding incidence rates of SARS-CoV-2 infection according to cohorts were: 2.44 cases per 1000 person-months (confidence interval (CI) 95%: 2.40–2.50) and 4.29 (95% CI: 4.15–4.43) for comparison cohort. The incidence rate ratio (IRR) of SARS-CoV-2 was 1.76 (95% CI: 1.69–1.83). When analyzing according to pregnancy period, the IRRs were 1.30 (95% CI: 11.20–1.41) during the puerperium and 1.19 (95% CI: 41.15–1.23) after pregnancy. In addition to pregnancy itself, other important risk factors were obesity (1.33 (95% CI: 1.23–1.44)) and diabetes (1.23 (95% CI: 11.00–1.50). Conclusion: Pregnant women are at increased risk of SARS-CoV-2 infection compared with women of childbearing age not pregnant. Nevertheless, there is a trend towards reverting during puerperium and after pregnancy.

## 1. Introduction

It has been suggested that the clinical outcome and consequences of SARS-CoV-2 infection in pregnant women might course differently from those in the general population; therefore, pregnancy has been considered a potential risk factor for COVID-19. During pregnancy, the maternal immune system has modulations that might affect the response to infections, as viruses [1,2]. For example, infection by other coronaviruses e.g., severe acute respiratory syndrome (SARS) and Middle Eastern during pregnancy has been associated with higher case fatality rates and more severe complications during pregnancy [3,4]. In addition to the systemic immunological changes during pregnancy, there are a decrease in lung capacity and inability to clear secretion that make them more vulnerable to respiratory infections [5]. Finally, during pregnancy there is an increase of thrombin production and an intravascular inflammation that results in a hypercoagulable state [6]. Thus, all these physiologic adaptations might trigger a severe course of pneumonia, with subsequent higher maternal and fetal morbidity and mortality [7,8].

Regarding evidence of COVID-19 and pregnancy, several metanalyses have been published, although some of them included small numbers of patients. Di Mascio et al. [9] concluded that hospitalized pregnant women with COVID-19 infection had higher rates of preterm birth, preeclampsia, cesarean delivery, and perinatal death. A more recent meta-analysis also concluded that severe SARS-CoV-2 infection was strongly associated with preeclampsia and other adverse maternal and neonatal outcomes [10]. Another meta-analysis focused on the clinical course of COVID-19 suggests that the clinical course and characteristics of COVID-19 in pregnant women seem similar to non-pregnant women [11]. Authors also reported how symptoms typically related to the virus, i.e., fever and cough followed by symptoms of anosmia, ageusia, myalgia, fatigue, sore throat, malaise, rigor, headache, and poor appetite were the most frequent observed in pregnant women [12].

It should be noted that the majority of studies selected either hospitalized women or symptomatic ones, overrepresenting this population. However, a high proportion of asymptomatic women have been found to be positive from routine nasal/throat swab PCR testing when admitting to give birth [13], and prior studies have suggested a more asymptomatic course of the disease in pregnant women [14,15]. Although it has been reported higher infection rates among pregnant women compared with non-pregnant women, [16,17] data remains controversial, and it is still too soon to conclude if pregnancy confers more susceptibility for SARS-CoV-2 infection. 

In order to widening the spectrum and representation of all SARS-CoV-2 cases in pregnant women, we aimed to find out whether the risk of SARS-CoV-2 infection in pregnant women differs from women of childbearing age but without pregnancy. We used data from the BIFAP database which contains systematically recorded data on more than ten million primary care patients in Spain.

## 2. Material and Methods 

### 2.1. Data Source 

BIFAP, an electronic medical record database with longitudinal population-based of anonymized electronic medical records of primary care practitioners and pediatricians (PCP), was used to conduct the current study. This database includes information of primary care physicians and pediatricians, including: demographic factors, consultation visits, referrals, hospital admissions, laboratory test results, diagnostic procedures, diagnoses, and prescriptions. At the time of the study, the database included information from nine participating Autonomous Regions (out of 17) in Spain. The distributions of age and sex are comparable with the Spanish population [18,19]. Clinical data are entered using international Classification of Primary Care—Second Edition (ICPC-2) and ICD-9. [20,21] and medications are automatically recorded by the PCP or specialists or during admissions to hospital, using the ATC classification [22]. Further details have been published elsewhere [23]. 

### 2.2. Source Population

The study population consisted of all women of childbearing age (15–49 years) during the study period between January 2020 and December 2020 (the latest date of the criteria was the date of entry into the study) from five regions with SARS-CoV-2 data available upon the date of conducting the study (up to 30 June 2021 for four regions and 31 December 2020 for one region, representing approximately 75–80% of BIFAP database). In order to participate in the study, and as an inclusion criterion, women must have been registered with their primary care physician at least one year before entering the study. This criterion and time frame (i.e., one year) serve to ensure a minimum of information recorded on the patients, and to be able to collect demographic data (lifestyle, such as BMI) and comorbidities, recorded by the physician beforehand. In addition, it was a prerequisite to ensure one year of medical history to be able to identify the gestational age in the cohort of pregnant women. All women with a diagnosis of SARS-CoV-2 before the study entry were excluded. 

### 2.3. Identification of the Cohort of Pregnant Women, Gestation Time and Comparison Cohort 

Once the cohort of women meeting the inclusion/exclusion criteria was constituted, the following indicators of pregnancy were identified: (i) indicators of conception (last menstrual period date record-LMP), (ii) indicators of end of pregnancy, which included: record of delivery, miscarriage, termination of pregnancy, ectopic pregnancy, planned abortion and/or intrauterine fetal death; and (iii) other codes compatible with a pregnancy, such as pregnancy test, prenatal visits, pregnancy complications, etc. After assignment of the validated gestational age, women identified as pregnant were classified according to pregnancy outcome into: (i) term pregnancy, (ii) miscarriage or (iii) stillbirth, (iv) unspecific pregnancy. All those women whose gestational age could not be calculated (pregnancy with non-specific gestational age) were excluded. The details of the methodology for pregnancy identification have been previously described by the members of this team using the BIFAP database [24]. For each pregnant woman, we matched 1:4 to non-pregnant women on the LMP date/matched date with the same age, region and follow-up time/length of pregnancy (Figure 1).

### 2.4. Follow–Up and Outcome Identification

From the LMP date/matched date we followed up both cohorts until the earliest of the following endpoints: a recorded diagnosis of SARS-CoV-2, death, transferred out from the database or the end of follow-up (31 June 2021). A patient was designated as a confirmed SARS-CoV-2 case if they meet one of the following criteria: a confirmed case of SARS-CoV-2 infection from the active surveillance system implemented during the SARS-CoV-2 pandemic and from hospital data or from intensive care unit (ICU). SARS-CoV 2 infection was confirmed by a positive result of RT-PCR test or positive antibody serology test. Of note, at the time of the study, there was not implemented other techniques to confirm SARS-CoV-2 infection such as antigen test.

### 2.5. Covariables 

We collected information on demographics and lifestyle factors, health care use, comorbidities and drug utilization. Information on comorbidities was collected any time before the LMP date/matched date, and health care utilization (measured as the number of GP visits in the year before the index date) was established in the year before the LMP date/matched date. Use of medications was identified in the three months prior to the LMP date/matched date, use was defined as having at least one prescription within that time window. Specific symptoms associated with SARS-CoV-2 infection such as anosmia, respiratory symptoms, digestive symptoms, fever and others were collected within the +/−7 days following SARS-CoV-2-recorded diagnosis. In addition, we looked for clinical complications of COVID-19 within the 28 days following the recorded infection. Specifically, we looked for pneumonia onset, bronchitis/bronchiolitis, thrombosis, valvopathy, and coagulopathy disseminated disease. 

### 2.6. Statistical Analysis

We conducted a descriptive analysis with categorical data presented using frequency counts and percentages, and continuous data using means with SD. Incidence rates (IR) and incidence rate ratio (IRR) of SARS-CoV-2 infection were calculated by type of cohort applying different time windows: during pregnancy, during puerperium defined as 42 days following end of pregnancy and from the end of puerperium up to the end of study period June 2021. Kaplan–Meier survival curves for SARS-CoV-2 Infection were calculated by type of cohort within each specific time window. Hazard ratios (HRs) and their 95% confidence intervals (CIs) were calculated using Cox proportional hazard models adjusted for age, obesity, hypertension, diabetes, asthma, COPD, cancer, multiple sclerosis, rheumatoid arthritis and falciform anemia. A two-sided *p* value < 0.05 was considered to be statistically significant. Statistical analyses were performed using the Stata package version 12.0 (StataCorp LP, College Station, TX, USA). 

## 3. Results

### 3.1. SAR-CoV-2 Infection Onset among Both Cohorts

The pregnancy cohort encompassed a total of 103,185 pregnancies matched to 412,740 women by age, region and follow up. The mean age of participating women was 32.3 years (median 33, SD 6.0). The distribution of pregnancy events was 81.3% birth, 0.3% stillbirths, 16.7% pregnancy losses, and 1% ectopic pregnancies. A total of 8.3% of pregnancies had a recorded diagnosis of SAR-CoV-2 infection and 6.0% among the comparison cohort. We then subdivided the SAR-CoV-2 infection diagnosis according to time of infection within the following time windows: (i) during pregnancy, (ii) during puerperium (42 days following the end of pregnancy) and (iii) after puerperium (from the end of puerperium until the end of study period (30th June 2021)). There were a total of 3522 pregnancies with a SARS-CoV-2 infection within the pregnancy and 8063 in the comparison cohort. The corresponding incidence rates (IRs) were 4.29 (95% CI: 4.15–4.43) cases per 1000 persons-months and 2.44 (95% CI: 2.40–2.50) cases per 1000 persons-months. Figure 2 shows the Kaplan–Meier distribution of SAR-CoV-2 infection with a log rank < 0.0001. The IRR of SARS-CoV-2 infection yielded an estimate of 1.76 (95% CI: 1.69–1.83). Figure 3 shows the Kaplan–Meier curves of cumulative incidence of SARS-CoV-2 infection by each time window. As seen in both figures, the pregnancy cohort had a higher incidence of SARS-CoV-2 infection (log rank tests < 0.0001).

Table 1 shows the incidence of SAR-CoV-2 infection according to each moment of risk. During the pregnancy window, the cohort of pregnancies had an IRR of 1.76 (95% CI: 1.69–1.83), this risk decreased during the puerperium 1.30 (95% CI: 1.20–1.41) and decreased a bit more after puerperium 1.19 (95% CI: 1.15–1.23).

### 3.2. COVID-19 Onset According with Pregnancy Event 

When we evaluated the incidence of SAR-CoV-2 infection according to pregnancy event, the corresponding incidence rates were: 4.19 cases (95% CI: 4.05–4.34) per 1000 person-months for women who gave birth (4.05–4.34), 5.29 for stillbirth (95% CI: 3.00–9.31), 5.81 for pregnancy loss (95% CI: 5.12–6.59) and 9.13 for ectopic pregnancy (95% CI: 5.68–14.69). Therefore, compared with women at childbearing age, their incidence rate ratios of SAR-CoV-2 infection based on pregnancy outcome were: 1.67 (95% CI: 1.60–1.74), 2.29 (95% CI: 1.30–4.03), 4.36 (95% CI: 3.82–12.43) and 7.70 (95% CI: 4.77–4.99) (Figure 4).

### 3.3. Clinical Course of SARS-CoV-2 Infection

We collected the most frequent symptoms and complications derived from SARS-CoV-2 infection. Acute symptoms were collected within the +/−7 days and complications from COVID-19 diagnosis up to 28 days after. Data are extremely limited, with an important underrecording of symptoms < 0.1% of prevalence among both cohorts. In terms of complications, pneumonia was the most frequent one (49 cases in the comparison cohort and 19 in the pregnancy cohort). There was none deceased in any group (Appendix A).

### 3.4. Cox Regression Analysis: Risk Factors for SARS-CoV-2 Infection

The following results are restricted to those for women with infection during pregnancy/matched period but not during puerperium or after puerperium. Frequency distributions of characteristics of the two study cohorts at the start of follow-up are shown in Table 2 and Table 3 for demographics, lifestyle factors, health care use, morbidities and medications. Pregnancy also showed to be a risk factor for SARS-CoV-2 infection (1.76 (95% CI: 1.69–1.83)) (Table 2).

Overall, the HRs associated with known risk factors for SARS-CoV-2 infection were: 1.23 (95% CI: 1.23–1.43) for diabetes, 1.33 (95% CI: 1.23–1.44) for obesity, 1.11 (95% CI 0.98–1.26) for hypertension and 1.28 (95% CI: 1.01–1.61) for cancer. Comorbidities such as asthma (1.01 (95% CI: 0.94–1.08)) or COPD (1.26 (95% CI: 0.70–2.24)) did not show a statistically significant association however, women receiving at least one prescription of corticosteroids (1.18 (95% CI: 1.00–1.39)) or drugs indicated for respiratory problems (1.07 (95% CI: 0.99–1.14) has a slightly increased risk of SARS-CoV-2 infection. Cardiovascular conditions did not show an increased risk of SARS-CoV-2 infection: corresponding estimates were 1.39 (95% CI: 0.72–2.67) for stroke and 1.66 (95% CI: 0.62–4.42) for TIA and 1.23 (0.88–1.72) for ischemic heart diseases, although numbers were small.

Statins showed an increased risk of SARS-CoV-2 infection (HR: 1.42 (95% CI: 1.11–1.82)). Drugs indicated for analgesia such as NSAIDs (1.11 (95% CI: 1.05–1.18)), paracetamol (1.18 (95% CI: 1.09–1.27)), opioids (1.15 (95% CI: 1.03–1.28)) and antimigraine (1.14 (95% CI: 1.04–1.25)) were slightly associated with an increased risk of SARS-CoV-2 infection. Last, users of PPIs presented a HR of 1.30 (95% CI: 1.19–1.42), however this result should be interpreted with caution due to confounding by indication (Table 3). 

NSAIDs: Non-steroidal anti-inflammatory drugs; ARBs: Angiotensin receptor blockers; ACEIS: Angiotensin-converting-enzyme inhibitors; H2 antagonists: Histamine H2 Antagonist; BZD: Benzodiazepines; SSRIs: Selective serotonin reuptake inhibitors

## 4. Discussion

Our study encompassed a total of 103,185 pregnant women matched with 412,740 women of childbearing age by age, LMP date/matched date, and length of pregnancy and region. Pregnant women had an increased risk of SARS-CoV-2 infection of 76%. We observed a trend towards a decreased risk according to time since pregnancy: During puerperium, the increased risk of COVID-19 was 30%, and it was 19% after puerperium, becoming almost the same after the puerperium with respect to the non-pregnant cohort. Prior reports have observed a higher prevalence of SARS-CoV-2 infection among pregnant women than the expected values initially [10]. Some reasons could be more frequent health care visits compared with non-pregnant women, but also a more intensified screening and detection of SARS-CoV-2. In fact, initially back in 2020, some hospitals started implementing routine COVID-19 upon admission to labor and delivery (L&D) and found a 20% prevalence of SARS-COV-2 infections with a high proportion of asymptomatic cases [25,26]. This approach that has become routine in clinical practice has served to accurately monitor women and provide resources appropriately. Current guidelines on key considerations regarding the management of COVID-19 in pregnancy include counseling about the increased risk for severe disease from SARS-CoV-2 infection and recommendations to protect themselves [27], attending routine antenatal care, testing symptoms, and getting vaccinated among others. In our study, and restricted to a pregnant cohort, women with a pregnancy loss had a higher risk of SARS-CoV-2 infection compared with women who gave birth, which suggests that the results cannot be explained via only admission to L&D. Among risk factors associated with early fetal loss are several inflammatory events, including systemic inflammation [2], that could involve the placenta. This fact could be worsening with SARS-CoV-2 infection as it has been described that it can provoke inflammation and placental insufficiency triggering the risk of fatal outcomes [2,28]. Results from meta-analyses identified an increased risk of abortion in mothers with a positive test result of SARS-CoV-2 although this evidence still remains inconclusive [29].

In terms of the clinical course of COVID-19, the vast majority of the study focused on risk factors associated with severity and fatal outcome rather than infection. Results from systematic reviews and meta-analysis suggest [30,31,32] that pregnant women with COVID-19 attending or admitted to hospitals are less likely to present symptoms such as fever, dyspnea, and myalgia and are more likely to be admitted to the intensive care unit and require invasive ventilation compared with non-pregnant women. In our study, we were not able to describe the clinical course as we found an extreme low proportion of recorded symptoms. Although numbers were very small, we found the same proportions of admissions to ICU, pneumonia, and thrombosis. However, we found a higher proportion of admissions to hospital in pregnant women, although the vast majority arrived within the third trimester (>95%), which indicates admissions to L&D and therefore checking SARS-CoV-2 status.

When we evaluated health conditions that can increase susceptibility to SARS-CoV-2 infection, pregnancy status was the most important risk factor followed by underlying conditions such as obesity and diabetes. In addition to changes in the immune system, there are also mechanical changes produced by the gravid uterus that can raise the diaphragm and produce physiological alterations in the shape of the lungs that can affect lung function. Pregnant women, apart from the anxiety due to pregnancy itself, might be more aware about the outcome of pregnancy and fetal status, which might create a more vulnerable state for viral infection. [33] Obesity leads to more adipose tissue and more angiotensin- converting enzyme 2 (ACE2) receptors on the cell surface, in which SARS-CoV-2 binds and penetrates in the cell [34]. Prior studies have also found a positive correlation among prenatal BMI and COVID-19 infection [35]. Likewise, diabetes involves a chronic inflammatory effect together with a prothrombotic stage that might play a worsen response to infections [36,37]. Having a history of cancer showed an increase in SARS-CoV-2 infection of 28%, these patients already share associated health conditions such as cardiac disease, diabetes, dyslipidemia, hypertension, obesity, osteoporosis, and osteopenia together with the immunocompromised state that infer an increase risk [38,39,40,41]. Interestingly, we did not find any association with established risk factors such as hypertension, asthma, and conditions related with the immune system such as multiple sclerosis or rheumatoid arthritis, although we should keep in mind the average age (32 years old) and relatively healthy status of this population; it should be noted that those conditions have been linked with disease severity rather than an increase in the likelihood of SARS-CoV-2 infection. Socioeconomic risk factors such as low income, specific living conditions (i.e., size, household composition), and social deprivation have been associated with COVID-19 incidence [42]. However, we could not collect these factors within the current data.

We used data from BIFAP which is representative of the Spanish population with respect to age, sex, and geographical region [43]. In Spain, the PCP is the gatekeeper to the health care system. According to the last National Health survey, a total of 98% of all citizens visited their PCPs at least once during 2017 [44]. Thus, the PCP represents not only the first visit to monitor pregnancy but also SARS-CoV-2 infection. In relation to pregnancy, prenatal care is also delivered by midwives, specialists, and hospitals, this results in any case in a small proportion of missed pregnancies considering the universal health care system offered in Spain. In relation to SARS-CoV-2 infection, we might have underreported some outcomes that occurred outside the GP surgery; however, all regions participating in BIFAP delivered COVID-19 data to BIFAP. It should be mentioned the small prevalence of clinical course symptoms such as cough, anosmia, fever, and chills. At the time of recording data (2020 and 2021 years), there was an overload in the Spanish National Health System that could have led to a decrease in recording information into the medical electronic system. As a result, we could not answer if the clinical course of COVID-19 differs across groups together with the proportion of asymptomatic patients.

Likewise, absent other electronic medical records, we could not assume that prescription or dispensing reflect actual drug intake; thus, some degree of misclassification cannot be ruled out. We might miss some information recorded outside the PCP surgery but in any case, it would be minimal and non-differential. Lastly, there is some degree of missing data, especially regarding the lifestyle factors in BIFAP. Nevertheless, according to prior publications using BIFAP, no differences in risk estimates were found when applying different strategies for controlling missing data [45,46].

## 5. Conclusions

In our study, we found that there was a higher risk of SARS-CoV-2 infection in pregnant women than in women without pregnancies. However, this increased risk tends to become more similar to non-pregnant women as the pregnancy progresses. Women with fatal pregnancy outcomes presented a higher risk of SARS-CoV-2 infection; thus tight monitoring at early stages of pregnancy might be crucial. Further studies are warranted in order to evaluate clinical course and detailed hospital information.

## Figures and Tables

**Figure 1 healthcare-10-02429-f001:**
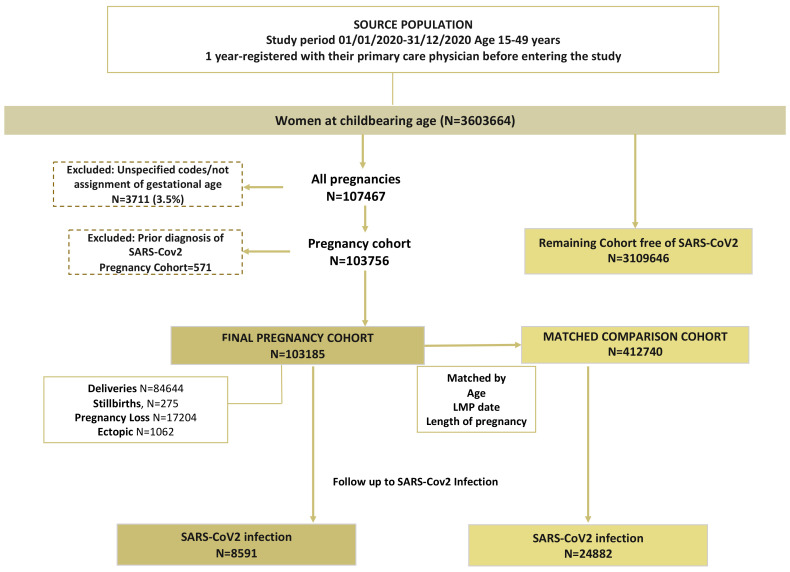
Flow chart of the study design.

**Figure 2 healthcare-10-02429-f002:**
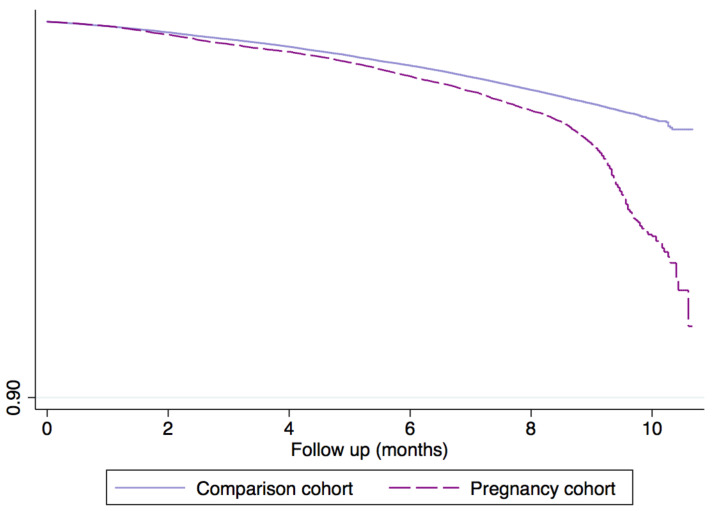
Kaplan–Meier survival estimate showing time to diagnosis of SARS-CoV-2.

**Figure 3 healthcare-10-02429-f003:**
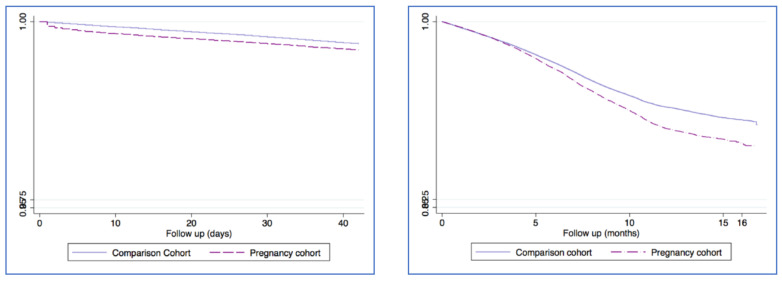
Kaplan Meier Cumulative incidence of SARS-CoV-2 infection according to puerperium period (figure in the left side) and post puerpuerium (figure in the right side).

**Figure 4 healthcare-10-02429-f004:**
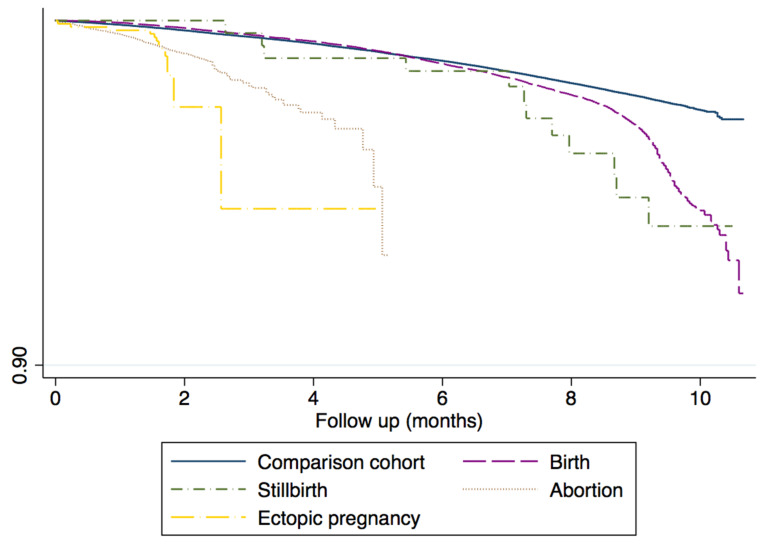
Kaplan Meier Cumulative incidence of SARS-CoV-2 infection according to outcome of pregnancy.

**Table 1 healthcare-10-02429-t001:** Incidence rates of COVID restricted to women during pregnancy and their pairs (i.e., excluding women infected during puerperium or after pregnancy).

Characteristics	Comparison Cohort N = 412,740	Pregnancy Cohort N = 103,185
During pregnancy		
Cases, n	8063	3522
Total person-months	3,291,191	821,374
Months (median)	9.3 (8.7–9.4)	9.3 (8.6–9.4)
Incidence rate per 100 0person-ye (95% CI)	2.44 (2.40–2.50)	4.29 (4.15–4.43)
Incidence Rate Ratio (IRR) 95% CI	-	1.76(1.69–1.83)
Puerperium		
Cases, n	2308	739
Total, person-months	542,756	133,495
Median months	1.4 (1.4–1.4)	1.4 (1.4–1.4)
Incidence rate per 100 0person-ye (95% CI)	4.25 (4.08–4.43)	5.54 (5.15–5.95)
Incidence Rate Ratio (IRR) 95% CI	-	1.30 (1.20–1.41)
After pregnancy	370,273	91,244
Cases, n	14,511	4330
Total, person-months	2,542,165	634,852
Median months	5.9 (2.2–9.6)	6 (2.3–9.7)
Incidence rate per 100 0person-ye (95% CI)	5.71 (5.62–5.80)	6.82 (6.62–7.03)
Incidence Rate Ratio (IRR) 95% CI	-	1.19 (1.15–1.23)

**Table 2 healthcare-10-02429-t002:** Baseline characteristics and its association with COVID onset.

	Non COVID N = 504,340		COVID N = 11,585		
Comorbidities	N	%	N	%	
Pregnancies	99,663	19.8	3522	30.4	1.76 (1.69–1.83)
Obesity	25,512	5.1	760	6.6	1.33 (1.23–1.44)
Diabetes	3615	0.7	96	0.8	1.23 (1.23–1.43)
HTA	10,092	2.0	252	2.2	1.11 (0.98–1.26)
Asthma	36,966	7.3	883	7.6	1.01 (0.94–1.08)
COPD	417	0.1	11	0.1	1.26 (0.70–2.24)
Arrhythmia	10,728	2.1	269	2.3	1.09 (0.97–1.23)
IHD	1197	0.2	34	0.3	1.23 (0.88–1.72)
Hypothyroidism	21,427	4.2	578	5.0	1.14 (1.05–1.24)
Depression	23,779	4.7	565	4.9	1.09 (0.99–1.18)
Dyslipidemia	24,602	4.9	610	5.3	1.11 (1.02–1.20)
HIV	913	0.2	13	0.1	1.42 (0.89–2.25))
Anemia year prior	3850	0.8	100	0.9	1.09 (0.89–1.33)
Psoriasis	509	0.1	6	0.1	0.51 (0.23–1.14)
Migraine	19,605	3.9	517	4.5	1.14 (1.04–1.25)
Epilepsy	3309	0.7	78	0.7	1.06 (0.85–1.32)
Gastritis	25,089	5.0	626	5.4	1.07 (0.99–1.16)
Celiac	2206	0.4	57	0.5	1.13 (0.87–1.47)
Rheumatoid Arthritis	963	0.2	22	0.2	1.02 (0.67–1.55)
Multiple Sclerosis	1076	0.2	19	0.2	0.84 (0.54–1.33)
Cancer	2608	0.5	71	0.6	1.28 (1.01–1.61)
Stroke	306	0.1	9	0.1	1.39 (0.72–2.67)
TIA	99	0.0	4	0.0	1.66 (0.62–4.42)
Valvopathy	673	0.1	14	0.1	0.95 (0.56–1.60)

Model adjusted by age, type of cohort, obesity, diabetes, hypertension, cancer, asthma, COPD, falciform anemia, multiple sclerosis and rheumatoid arthritis. HTA: Hypertension; COPD: Chronic obstructive pulmonary diseases; IHD: Ischemic heart disease; HIV: Human immunodeficiency virus; TIA: Transient ischemic attack.

**Table 3 healthcare-10-02429-t003:** Baseline characteristics and its association with COVID onset.

	Non COVID N = 504,340		COVID N = 11,585		
Treatment Pre-Pregnancy	N	%	N	%	
NSAIDs	51,238	10.2	1334	11.5	1.11 (1.05–1.18)
Heparin	1380	0.3	43	0.4	1.24 (0.92–1.68)
Antihypertensives	175	0.0	6	0.1	1.27 (0.57–2.84)
Diuretics	829	0.2	12	0.1	0.65 (0.37–1.15)
Calcium antagonists	584	0.1	14	0.1	1.05 (0.62–1.80)
ARBs	1326	0.3	26	0.2	0.89 (0.60–1.32)
ACEIs	1607	0.3	30	0.3	0.79 (0.55–1.15)
Antiplatelets	1619	0.3	50	0.4	1.26 (0.96–1.67)
Aspirin	1584	0.3	50	0.4	1.29 (0.98–1.71)
Beta Blockers	1861	0.4	43	0.4	1.07 (0.79–1.44)
PPIs	17,502	3.5	508	4.4	1.30 (1.19–1.42)
H2 ANTAGONISTS	1345	0.3	21	0.2	0.59 (0.38–0.90)
Antacids	67	0.0	1	0.0	0.67 (0.09–4.77)
Vit K Antagonists	333	0.1	5	0.0	0.75 (0.31–1.81)
Phenytoin	30	0.0	3	0.0	4.34 (1.40–13.46)
Valproic acid	719	0.1	18	0.2	1.22 (0.77–1.94)
Antibiotics	38,542	7.6	1059	9.1	1.11 (1.04–1.19)
Respiratory drugs	35,742	7.1	895	7.7	1.07 (0.99–1.14)
Opioids	12,380	2.5	337	2.9	1.15 (1.03–1.28)
Migraines	4009	0.8	100	0.9	1.14 (1.04–1.25)
Antiepileptics	7202	1.4	159	1.4	1.04 (0.89–1.22)
Anxiolytics	21,977	4.4	495	4.3	1.04 (0.95–1.14)
Allergy	27,337	5.4	671	5.8	1.04 (0.96–1.12)
BZD	22,827	4.5	523	4.5	1.07 (0.97–1.16)
Antidepressants	14,054	2.8	332	2.9	1.10 (0.99–1.23)
SSRIs	11,939	2.4	279	2.4	1.09 (0.97–1.23)
Insulin	1593	0.3	37	0.3	0.83 (0.55–1.23)
Oral antidiabetics	2078	0.4	64	0.6	1.21 (0.93–1.58)
Statins	2226	0.4	66	0.6	1.42 (1.11–1.82)
Paracetamol	22,448	4.5	637	5.5	1.18 (1.09–1.27)
Corticosteroids	5163	1.0	146	1.3	1.18 (1.00–1.39)
Thyroid hormones	14,312	2.8	354	3.1	1.04 (0.93–1.15)

Model adjusted by age, type of cohort, obesity, diabetes, hypertension, cancer, asthma, COPD, falciform anemia, multiple sclerosis and rheumatoid arthritis.

## Data Availability

The data that support the findings of this study are available from the corresponding author upon reasonable request. Some data may not be made available because of privacy or ethical restrictions. Cea Soriano is the guarantor of this work and, as such, had full access to all the data in the study and takes responsibility for the integrity of the data and the accuracy of the data analysis.

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
