# Peer review of "The Risk of SARS-CoV-2 Infection in Pregnant Women: An Observational Cohort Study Using the BIFAP Database"

_healthcare, 2022, doi:10.3390/healthcare10122429_

Round 1

Reviewer 1 Report

The manuscript present a study conducted in BIFAP, population-based database in Spain, including information of patients attending primary care and used for epidemiologic research. This was an analysis to estimate incidence of SARS-CoV-2 infection in a large cohort of pregnant women identify by a validated algorithm. The authors found that pregnant women were at increased risk of infection compared to a matched cohort of nonpregnant women. 

Please find below my comments to the manuscript.

Introduction:

 Line 40-42. The authors justify the need of this new study as others only included hospital or symptomatic cases out of the whole spectrum of SARS-CoV-2 infection, do they mean overrepresentation of only “severe infection” and not representative of all SARS-CoV-2 ?? Using BIFAP database they believe the representativeness will improve ? 

Methods. 

Line 81-22: Descriptions on validity and representativeness should be referred to the subgroups of 5 regions with sufficient info that were finally used in this study.

Line 98-99, Women with prior SARS-CoV-2  infection are not included, authors may mention that this study only covers 1st ever infection. Please define this exclusion criteria clearly: e.g.  as any test + or a confirmed diagnosis ? 

A clear definition of criteria to consider a confirmed case with SARS-CoV-2 infection is needed. Authors include cases confirmed by hospital and UCI, and those included in the registry. What type of registry is this?  What are the criteria to be included in that registry?  A symptomatic infections without PCR+ will be considered ? or an asymptomatic patient with  PCR + or antigen test + , ? Or both ?? Please clarify. 

Line 130: How many pregnancies were included as “unspecific pregnancies” ? A brief description/ definition  of this group is necessary. Where these women finally included in the analyses ? 

It may be useful to see a chart with all initially identified women, and how many were excluded at each step and the reason for the exclusion,  and how many finally included in the analyses and the distribution of according to pregnancy outcome. 

Line 118. “From LMP date we followed up both cohorts ….”  this first sentence is confusing , reader may interpret that LMP date was known for both cohorts ( pregnant and non-pregnant women, and follow-up started at this point. Line 118 , line 126 it should be amended and clarify  “LMP/matched date” as the start date to follow-up

Line 127. Why healthcare use measured with respect to index date and prescription with the respect  to start to follow-up date ?  

What is the index date? It is the same as LMP/matched date?? Or is the date of SARS-CoV-2 infection ?  This is an important issue that need to be clear and named the dates correctly.

Results

Presentation of KM graphs are very intuitive to see the risk during time follow-up, but authors need to add some description or comments of the meaning of this figures to help the correct interpretation of results, not only the title of the figure.

Figure 3, suggest to drop it, as the data is in text, and also it could be added as extra lines in table 1, for each pregnancy outcome.

Line 225 . Title of figure 4 does not correspond with the KM curve presented, please edited

Line 227. Title of fire 5, is not correct. No only comedication also comorbidity estimates are presented.    

Suggest to review some figures that could be more clear and informative if  presented in table , specifically figure 3 that could be added to table 1 the IRR and 95%CI . An figure 5 that is not clear to see at it is, and may be easier to the reader to see a table.  

Discussion 

Line 229. Matched by  LMP date ( date start pregnancy but not length of pregnancy 

Line 239.I believe the reason the see increased risk at the end of pregnancy is due to checking status at admission 

Line 261.Higher admission to ICU by be a reflection not of severity but an excess of care among pregnant women due to their condition. Suggest to discuss. 

References

Please review reference citations. Some references seems to be incomplete ( e.g. Ref 19 and 20) and need edition.  

Is Supplemental material needed ? It is not referred in the manuscript.

Author Response

response to reviewer is attached

Reviewer 2 Report

This study is a statistical study showing the risk of SARS-CoV2 infection in pregnant women. It was a very interesting paper, and I thought that this research was suitable for this paper.

 I have a few questions.

First, In reference 11, you state an opinion contrary to your research, but please consider this point.

Next, how do the authors think that pregnant women are infected?

If it's a droplet infection, do you think wearing a mask would have reduced the infection?

So, did you check the presence or absence of masks in this study?

If the probability of infection at an obstetric outpatient clinic is high, is it correct to assume that the probability of infection at an outpatient clinic of internal medicine is the same for women with complications?

How do you think about the number of days of obstetric visits and the number of infected people in the clinic in this study?

A final question, discuss what can be done to reduce the risk of infection for pregnant women.

Please explain the above by citing the literature in modified paper.

Author Response

response to reviewer is attached
